# Antibiofilm and anti-quorum sensing activity of *Psidium guajava* L. leaf extract: *In vitro* and *in silico* approach

**Mo Ahamad Khan**[1], Ismail Celik[2], Haris M. Khan[1], Mohammad Shahid[3], Anwar Shahzad[4], Sachin Kumar[1], Bilal Ahmed[5]*

**1** Department of Microbiology, Faculty of Medicine, Aligarh Muslim University, Aligarh, India, **2** Department of Pharmaceutical Chemistry, Faculty of Pharmacy, Erciyes University, Kayseri, Turkey, **3** Department of Microbiology, Immunology and Infectious Diseases, College of Medicine and Medical Science, Arabian Gulf University, Manama, Kingdom of Bahrain, **4** Department of Botany, Faculty of Life Science, Aligarh Muslim University, Aligarh, India, **5** Agricultural and Biological Engineering, Purdue University, West Lafayette, IN, United States of America

* ahmed249@purdue.edu

**Data Availability Statement:** All relevant data are within the paper and its Supporting Information files.

## Abstract

The quorum sensing mechanism relies on the detection and response to chemical signals, termed autoinducers, which regulate the synthesis of virulence factors including toxins, enzymes, and biofilms. Emerging therapeutic strategies for infection control encompass approaches that attenuate quorum-sensing systems. In this study, we evaluated the anti-bacterial, anti-quorum sensing, and anti-biofilm activities of *Psidium guajava* L. methanolic leaf extracts (PGME). Minimum Inhibitory Concentrations (MICs) of PGME were determined as 500 µg/ml for *C. violaceum* and 1000 µg/ml for *P. aeruginosa* PAO1. Significantly, even at sub-MIC concentrations, PGME exhibited noteworthy anti-quorum sensing properties, as evidenced by concentration-dependent inhibition of pigment production in *C. violaceum* 12742. Furthermore, PGME effectively suppressed quorum-sensing controlled virulence factors in *P. aeruginosa* PAO1, including biofilm formation, pyoverdin, pyocyanin, and rhamnolipid production, with concentration-dependent inhibitory effects. Phytochemical analysis utilizing GC-MS revealed the presence of compounds such as alpha-copaene, caryophyllene, and nerolidol. *In-silico* docking studies indicated a plausible mechanism for the observed anti-quorum sensing activity, involving favorable binding and interactions with QS-receptors, including RhlR, CviR', LasI, and LasR proteins. These interactions were found to potentially disrupt QS pathways through suppression of AHL production and receptor protein blockade. Collectively, our findings propose PGME as a promising candidate for the treatment of bacterial infections. Its attributes that mitigate biofilm development and impede quorum-sensing mechanisms highlight its potential therapeutic value.

## Introduction

The significance of using antimicrobials carefully to combat the growth of antimicrobial resistance in pathogenic microorganisms is crucial. The rapid increase in antibiotic resistance

**Funding:** The authors received no specific funding for this work.

**Competing interests:** NO authors have competing interests.

worldwide poses a significant health concern, leading to higher rates of illness and death. Bacterial pathogens adapt quickly to challenging environments, leading to multidrug resistance. In response, natural antimicrobial compounds from plants have become important alternatives to traditional antimicrobials and antibiotics. As a result, there is a pressing need to identify new drug targets and create novel treatments for addressing bacterial infections. Notably, secondary metabolites from plants, with their antimicrobial, anti-biofilm, and anti-quorum sensing properties, offer a promising approach to combat bacterial infections [1–3].

Quorum sensing (QS) represents a communication mechanism employed by bacteria to facilitate intercellular interactions. This intricate process serves as a regulatory pathway for the expression of vital attributes, including virulence factors, biofilm formation, and the production of secondary metabolites. Additionally, it plays a pivotal role in the adaptation to environmental stressors, encompassing diverse bacterial competition systems, such as secretion systems [4, 5]. This intricate orchestration of bacterial cells is scored through the secretion of specialized signaling molecules known as autoinducers. Among the extensively studied quorum sensing systems in Gram-negative bacteria, those reliant on N-acyl-homoserine lactones (AHLs) as signal molecules have taken the forefront [6, 7]. Research has convincingly demonstrated the indispensable role of bacterial quorum sensing systems in the coordinated secretion of virulence factors (VFs) and the assembly of biofilms [8]. Biofilms, intricate three-dimensional structures constituted by aggregations of bacterial cells enclosed within extracellular polymeric substances, are generated by the microorganisms themselves [9]. Notably, the advent of biofilm formation substantially curtails the susceptibility of bacteria to conventional antibacterial agents [10]. Epidemiological investigations have notably established biofilms as contributing factors in over 65% of nosocomial infections, accounting for up to 80% of human microbial infections [10–12].

Addressing bacterial pathogenicity, the utilization of quorum sensing inhibitors (QSIs) presents a strategic route by targeting the quorum sensing systems (QSS) rather than direct cell eradication. Because it has no effect on bacterial growth, this technique gives the attenuation of bacterial pathogenicity to be readily removed without pressuring the germs to acquire resistance [13]. This innovative approach holds the promise of mitigating or delaying the emergence of antibiotic resistance, thereby diminishing the reliance on conventional antibiotics. Fundamental to bacterial persistence, virulence factors and biofilm formation represent compelling targets. By impeding these essential processes, novel therapeutic modalities can be formulated that impose minimal selection pressure for resistance, distinguishing them from traditional strategies [14]. Among Gram-negative bacteria, such as *Pseudomonas aeruginosa*, a quartet of QSS—LasI/R, RhlI/R, PQS, and IQS—govern VF production.

Key players include LasI and RhlI, responsible for synthesizing N-(3-oxodecanoyl)-l-homoserine lactone (OdDHL) and N-butanoyl homoserine lactone (BHL) respectively. Concurrently, *PQS*, a quinolone synthase, yields 2-heptyl-3-hydroxy-4(1H) quinolone. Binding to cognate receptors, OdDHL associates with LasR while BHL interacts with RhlR, culminating in the formation of LasR-OdDHL and RhlR-BHL complexes. This molecular interaction cascade triggers the transcriptional activation of associated genes, such as *lasI* and *lasR*, thereby initiating autoinduction. This intricate network further governs the expression of genes steering virulence and biofilm formation, including elastase, alongside instigating *rhl* and *pqs* QS systems. Additionally, the RhlR-BHL complex catalyzes the biosynthesis of rhlI and various QS-regulated virulence factors such as pyocyanin and rhamnolipid [15, 16]. The PqsR-PQS complex orchestrates PQS-mediated gene cascades, overseeing the expression of pivotal virulence factors like pyocyanin and rhamnolipids [17]. In routine conditions, the IQS system is tightly controlled by Las, whereas in the event of phosphate depletion stress, the IQS system assumes control in lieu of the central las system [18].

Given the well-documented role of quorum sensing systems in orchestrating virulence factors and biofilm development among Gram-negative bacterial pathogens, strategic interventions directed at these systems hold promise as effective therapeutic avenues against resistant strains. The natural environment boasts a plethora of diverse bioactive compounds with potential utility. Phytochemicals derived from medicinal plants have demonstrated potency as effective anti-quorum sensing agents against an array of pathogenic bacteria.

Among these, *Psidium guajava* L. (guava), a member of the Myrtaceae family and widely distributed across India, has found application in traditional medicine. Guava leaf extracts have garnered attention for their multifaceted properties, including anticancer [19], antidiabetic [20], antidiarrheal [21], antimicrobial [22], antimutagenic [23] and antioxidant [24] effects. Initial investigations have indicated that guava leaf extract possesses the capacity to modulate the quorum sensing of *C. violaceum* [25]. In this context, the present study evaluated the corroborative anti-virulence and anti-biofilm activities of guava leaf extracts, shedding light on the validation of its traditional medicinal usage.

The ensuing exploration encompasses a comprehensive assessment of guava leaf extracts, gauging their antimicrobial efficacy, their potential to disrupt quorum sensing, and their capacity to counteract biofilm formation. In this work, we evaluated the antimicrobial, anti-quorum sensing and antibiofilm activities of guava leaf extracts against *Chromobacterium violaceum* 12472 and *Pseudomonas aeruginosa* PAO1. Specifically, we investigate the substantial reduction in biofilm formation, alongside the modulation of pyocyanin, pyoverdin, rhamnolipid, and bacterial motility. Employing molecular modeling techniques, we endeavor to elucidate the potential roles of phytoconstituents in impeding biofilm formation and downregulating quorum sensing-regulated violacein production.

## Materials and methods

### Bacterial strains and culture conditions

The study employed two bacterial strains, namely *Chromobacterium violaceum* 12472 and *Pseudomonas aeruginosa* PAO1. These strains were individually cultured and maintained in Luria-Bertani (LB) broth, each at specific temperature conditions. *C. violaceum* 12472 was cultivated at 28°C, while *P. aeruginosa* PAO1 was maintained at 37°C.

### Plant materials and extract preparation

*Psidium guajava* L. (Guava) leaves were meticulously collected from trees within the premises of the Aligarh Muslim University, Aligarh, India. These leaves underwent authentication at the Department of Botany, Aligarh Muslim University, Aligarh, India. After collection, the leaves were gently rinsed with tap water, dried under shaded conditions, and subsequently ground into a coarse powder using a blender. A 20% concentration was achieved by mixing 50 g of guava leaf powder with 250 mL of methanol. The resultant mixture was subjected to 24–36 hours of shaking at 150 rpm on a magnetic stirrer. Subsequent steps involved filtration using Whatman grade filter paper I, followed by centrifugation at 4000 rpm for 5 min at 25°C. The supernatant was then concentrated at 40°C under reduced pressure using a rotary evaporator. The plant extract was store at 4°C, as a stock solution with concentration of 8 mg/ml for further use.

### Antimicrobial screening and minimal inhibitory concentration (MIC) determination

The antibacterial potential of all extracted fractions was initially screened using the agar well diffusion method on Mueller Hinton agar plates. Subsequently, the methanolic extract of

*Psidium guajava* (PGME) exhibited notable antibacterial activity, making it the focus of further experimentation. The MIC of PGME was determined through a modified broth microdilution technique [26]. This entailed the use of a 96-well microtiter plate, wherein PGME was subjected to two-fold serial dilutions in Muller-Hinton broth to generate a concentration range (250–8000 μg/mL). After inoculating bacteria into varying concentrations of PGME and appropriate controls, incubation was conducted at 37˚C for 24 hours. The addition of TTC dye (3 mg/mL) and subsequent visual observation after 20–30 min facilitated the identification of growth, as evidenced by a change in color to pink.

## Time-kill assays

Time-kill assays were conducted within the confines of Mueller-Hinton Broth (MHB) medium, as documented in the work by Zainin [27]. The plant extracts were suitably diluted in MHB medium that containing inoculum, resulting in a series of final concentrations, 0-MIC (control), 1/2-MIC, 1-MIC, 2-MIC, and 4-MIC. These cultures (final volume of 1 mL), were subjected to incubation at 37˚C while agitating at a rate of 200 rpm. At predetermined time intervals (0, 2, 4, and 8 hours), aliquots of 100 μL were extracted and transferred to microcentrifuge tubes. Subsequently, the obtained samples underwent serial dilution at a ratio of 1:100 in 1% phosphate-buffered saline (PBS) before plating onto Mueller-Hinton Agar (MHA) plates. The colonies that emerged on the MHA plates following 24 hours of incubation at 37˚C were enumerated, and the colony-forming units per milliliter (CFU/ml) were determined. All experimental procedures were meticulously performed in triplicates, and the results were depicted in the form of a logarithmic plot illustrating the relationship between log CFU/ml and time (h).

## Violacein inhibition assay

To qualitatively assess the violacein inhibitory potential of PGME, the method outlined by Husain et al. [28] was employed. Specifically, LB soft agar (0.5% w/v agar) containing *C. violaceum* 12472 ($10^6$ CFU/mL) was overlaid onto LB agar plates, allowing for absorption. After absorption, wells were cut with sterile pipette tips and Each individual well received an inoculation of 50 microliters of plant extract, with the negative control consisting of DMSO (0.5% v/v). Following an incubation period of 24 hours at 37˚C, the presence of pigment inhibition around the disc signified positive QS inhibition, with results expressed in terms of pigment inhibition or growth inhibition diameter (mm).

## Quantification of violacein

The method of Blosser and Gray [29] was used to extract and quantify the violacein pigment produced by *C. violaceum* 12472. Specifically, 100 μL *C. violaceum* was inoculated with various PGME concentrations in 10 mL LB broth and incubated at 30˚C with agitation for 24 h. Post-incubation, the culture was centrifuged at 10,000 rpm for 5 min to precipitate the insoluble violacein, pellet was then suspended in DMSO and centrifuged at 10,000 rpm for 5 min to remove the bacterial cells. The resulting supernatants were subjected to violacein measurement at 585 nm.

## Biofilm inhibition assay

**Crystal violet method.** The impact of sub-MICs of PGME on biofilm development was assessed using a 96-well plate [30]. Overnight growth cultures ($>10^7$ CFU/mL) of the test strains were dispensed into the 96-well MTP with 150 μL of LB medium, containing sub-MIC

levels of PGME or without, and incubated at 37˚C for 24 h. After incubation, PBS was used to wash the MTP wells to eliminate unattached cells. Crystal violet solution (0.1% w/v) was then added to visualize the biofilm, which adhered to the EPS and bacterial cells within it. The unbound crystal violet was removed through rinsing, and ethanol was used to solubilize the biofilm bound CV. The absorbance was measured at $OD_{620}$ using spectrophotometry.

**Scanning electron microscopy (SEM) of biofilm.** Biofilms of the test bacteria were cultured in LB broth on 1 $cm^2$ coverslips within a 12-well microtitre plate. In the presence and absence of sub-MICs of guava leaf extracts, mid-log grown bacterial cultures (100 μL) were introduced, supplemented with the highest sub-MIC (MIC/2) of PGME. Following a 24-hour incubation period, coverslips were rinsed with distilled water to remove any planktonic bacterial cells and air-dried. After thorough drying, the biofilms were fixed with 2.5% (v/v) glutaraldehyde for 2 hours. Gradual dehydration was achieved by immersing coverslips in graded ethanol solutions (30%, 50%, 70%, 90%, and 100%) for 5 min at room temperature. After a final wash with PBS, the coverslips were gold-coated and examined using a scanning electron microscope to visualize the biofilms.

**Assay for pyocyanin production.** The effect of sub-MICs of plant extract fractions on pyocyanin synthesis by *P. aeruginosa* PAO1 was assessed using the method established by Essar et al. [31]. Briefly, a mixture of 5 ml of thoroughly mixed supernatant (with and without test fractions) from an overnight culture in Pseudomonas Broth (PB) was combined with 3 ml of chloroform and vortexed for 5 min. Subsequently, pyocyanin was re-extracted from the organic phase using 1 ml of 0.2 N HCl. The absorbance was measured at 520 nm against 0.2 N HCl as a blank. The presence of various shades of pink to red color in the acid phase indicated the presence of pyocyanin. Results were expressed as the percentage inhibition of treated sets compared to control.

**Pyoverdin production assay.** For evaluating pyoverdin production, the test bacteria were cultured with and without sub-MICs of guava plant extracts at 37˚C. After 24 hours, the PAO1 culture was centrifuged to obtain the supernatant. Next, 100 μl of the supernatant was mixed with 900 μl of 50 mM Tris-HCl buffer (pH 7.4), and the resulting mixture was excited at 405 nm using a spectrofluorometer. The fluorescence intensity was measured at 465 nm [32].

**Assay for rhamnolipid activity.** Rhamnolipid activity was determined by extracting 300 μL of supernatant from both treated and untreated *P. aeruginosa* PAO1 cultures using 600 μL of diethyl ether, followed by vortexing for 1 min. After 30 min of phase separation, the organic phase (OP) was collected and dried through evaporation. The dried OP was solubilized by adding 100 μL of deionized water and gently shaking. This solution was then mixed with 900 μL of orcinol solution (0.19% Orcinol in 53% $H_2SO_4$) and heated at 80˚C for 30 min in a water bath. After cooling for 15 min, the resulting mixture was assessed [33].

**Exopolysaccharide (EPS) extraction.** *P. aeruginosa* PAO1 bacterial strain was grown with and without PGME for 24 hours at 37˚C. Following growth, the PAO1 culture was centrifuged to collect the supernatant. To precipitate EPS, the filtered supernatant was mixed with three volumes of chilled ethanol (100%) and allowed to precipitate overnight at 4˚C [34]. The quantity of precipitated EPS was determined by mixing 1 mL of the sample with 1 mL of cold phenol (5%) and 5 mL of concentrated $H_2SO_4$. The mixture was stirred until it turned red, and its intensity was measured at 490 nm using a spectrophotometer [35].

**Gas Chromatograph Mass Spectrometer (GC/MS).** GC/MS analysis of PGME was carried out using a GC/MS-Shimadzu QP-2010. Electron ionization was employed for GC/MS spectroscopy. Helium gas served as the mobile phase with a flow rate of 1 ml/min. The GC injector and MS detector were set to 260˚C. The oven temperature was initially set at 100˚C, gradually increased to 280˚C at a rate of 10˚C/min, held at 250˚C for 3 min, and eventually ramped to 250˚-280˚C at a rate of 30˚C/min, with a 2-min hold at 280˚C. Samples were diluted

with methanol. Compound identification was achieved by comparing mass spectra of detected peaks with a library (NIST library), based on peak area and retention time.

***In silico* studies: Homology modeling.** Due to the unavailability of the 3D structure of the *Pseudomonas aeruginosa* regulatory protein RhlR, the 3D structure obtained from UniProt with the code UniProtKB: P54292 was employed, estimated using the AlphaFold database (https://alphafold.ebi.ac.uk/entry/P54292). The suitability of the RhlR's 3D structure was assessed using the Ramachandran Plot obtained from the PROCHECK Server (https://saves. mbi.ucla.edu/).

**Molecular docking.** Molecular docking experiments were conducted using the CB-Dock server (http://clab.labshare.cn/cb-dock/php/). This process encompassed identifying binding sites, calculating their center and size, and tailoring the docking box dimensions based on query ligands. The actual molecular docking was performed using AutoDock Vina. The 3D structures of (-)-alpha-Copaene (PubChem CID: 442355), beta-Caryophyllene (PubChem CID: 5281515), and Nerolidol (PubChem CID: 5284507) compounds were sourced from the PubChem database. The 3D structures of targeted proteins, namely PDB ID: 3QP1 for CviR, PDB ID: 2UV0 for LasR, PDB ID: 1RO5 for LasI, and the forecasted structure from the Alpha-Fold Protein Structure Database through homology modeling for RhlR, were employed. To validate the docking studies, cocrystal ligands from the crystal structure of CviR and LasR proteins were employed for self-docking. The Root Mean Square Deviation (RMSD) values were calculated between cocrystal ligands and re-docking ligands. Binding poses and interaction diagrams were generated using BIOVIA Discovery Studio Visualizer v21.

**Statistical analysis.** The data provided in this study were reported as the mean of triplicate values, with the standard deviation (SD) indicated. Statistical analysis was conducted using the SPSS application software, and one-way ANOVA was employed to assess the disparities between test samples and controls. The significance of variations in various parameters, including violacein, pyocyanin, EPS, biofilm, and other virulence factors, was established through one-way ANOVA.

## Results and discussion

### Minimum inhibitory concentration (MIC)

The antibacterial activity of the methanolic extract of *Psidium guajava* (PGME) led to its selection for determining the MIC. For *C. violaceum*, the MIC value of PGME was established at 500 µg/ml, while for PAO1, it was found to be 1000 µg/ml. Subsequent experiments utilized sub-MIC concentrations of PGME, ranging from MIC/2 to MIC/16. Research investigating the antibacterial efficacy of guava leaf extracts has unveiled considerably high MIC values for *P. aeruginosa*. For instance, methanolic extracts derived from guava leaves exhibited an MIC of 250 mg/mL [36]. Similarly, Sanches et al. [37] show the MIC range for *P. aeruginosa* >1000µg/ml.

In order to enhance the precision of assessing the antibacterial efficacy of the plant extracts that exhibit noteworthy antibacterial and anti-quorum sensing activity (PGME) we conducted time-kill assays. These assays involved subjecting strains of *Pseudomonas aeruginosa* PAO1 and *C. violaceum* to various concentrations within the range of MICs spanning from 0-MIC to 4-MIC for the corresponding extracts. The outcomes from the experiments conducted with *P. aeruginosa* and *C. violaceum* are presented in **Fig 1**.

These findings suggest that the plant extracts hold promise as potential antimicrobial agents for treating bacterial infections caused by these microorganisms. Further investigations are needed to identify the specific bioactive compounds responsible for this antimicrobial activity and to evaluate the safety and efficacy of these extracts in clinical applications.

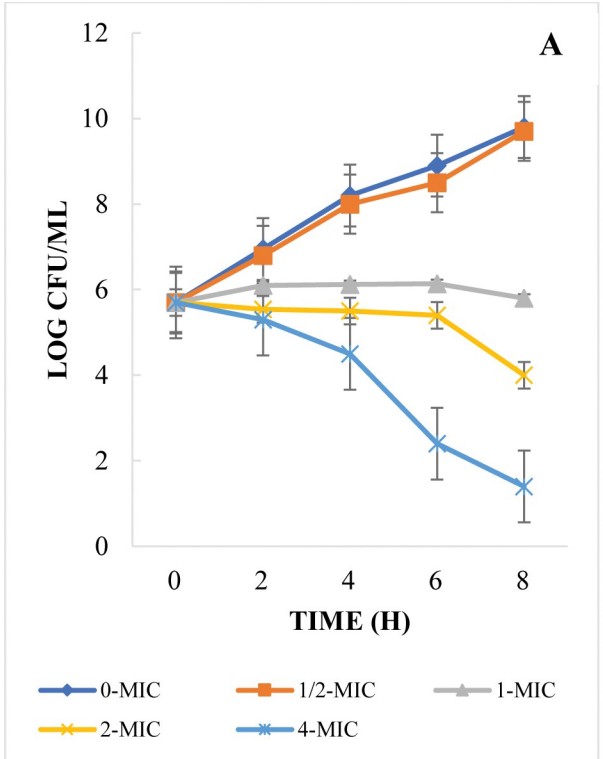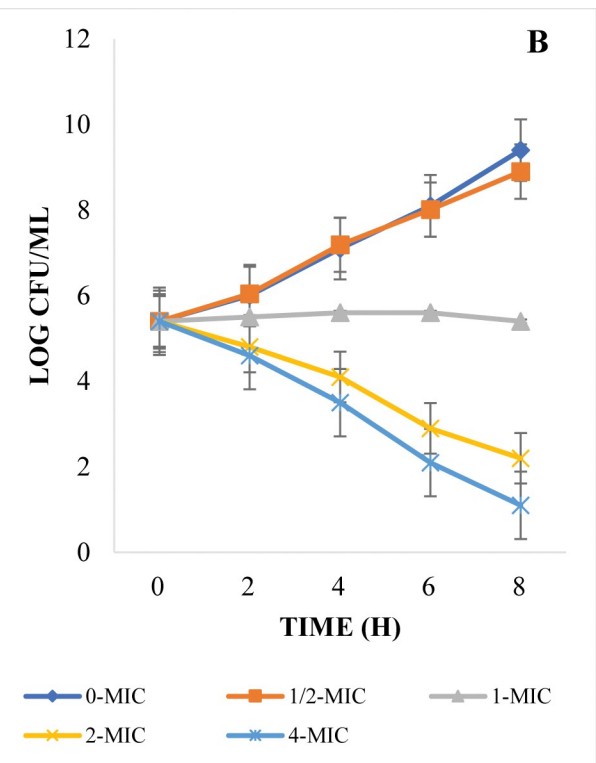

**Fig 1. The time kill assay plots showing the inhibition of *P. aeruginosa* PAO1 and *C. violaceum* by plant extracts (PGME) at 0 MIC, 1/2 MIC, 1-MIC, 2-MIC, and 4-MIC.** (A) The plot for *P. aeruginosa* PAO1 treated with PGME. (B) The plot for *C. violaceum* treated with PGME.

### Anti-quorum sensing activity

The preliminary assessment of anti-quorum sensing (QS) activity of guava extract was confirmed through the inhibition of violacein formation in *C. violaceum* 12472. Violacein, a pigment produced by *C. violaceum* in response to QS signaling, serves as an indicator of quorum sensing activity. As depicted in **Fig 2**, the non-purple zone on agar indicates anti-QS activity. This observation aligns with previous studies involving plant extracts like *Persicaria maculosa* and *Bistorta officinalis* that effectively inhibit violacein production [38, 39]. *Artemisia argyi* leaves extract also reduced violacein formation in *C. violaceum* 12472 [40].

### Violacein inhibition assay in *C. violaceum*

To validate the quorum sensing (QS) inhibitory potential of PGME, the extraction and quantification of violacein from *C. violaceum* 12472 culture was conducted. The presence of PGME at sub-MIC concentrations led to a concentration-dependent reduction in violacein synthesis, as depicted in **Fig 3**.

Notably, the highest sub-MIC concentration of PGME (250 μg/mL) demonstrated a substantial reduction of approximately 69.28% in violacein production. These results reinforce the notion that guava extract effectively inhibits QS-regulated violacein synthesis in *C. violaceum*.

The quantitative and qualitative analyses of the anti-QS effect of guava methanolic leaf extract on *C. violaceum* 12472 violacein production both underscore its concentration-dependent behavior. Comparable to the current study, a study by Cosa et al. demonstrated a 47.70% reduction in violacein pigment production by *Calpurnia aurea* (Aiton) Benth plant extract

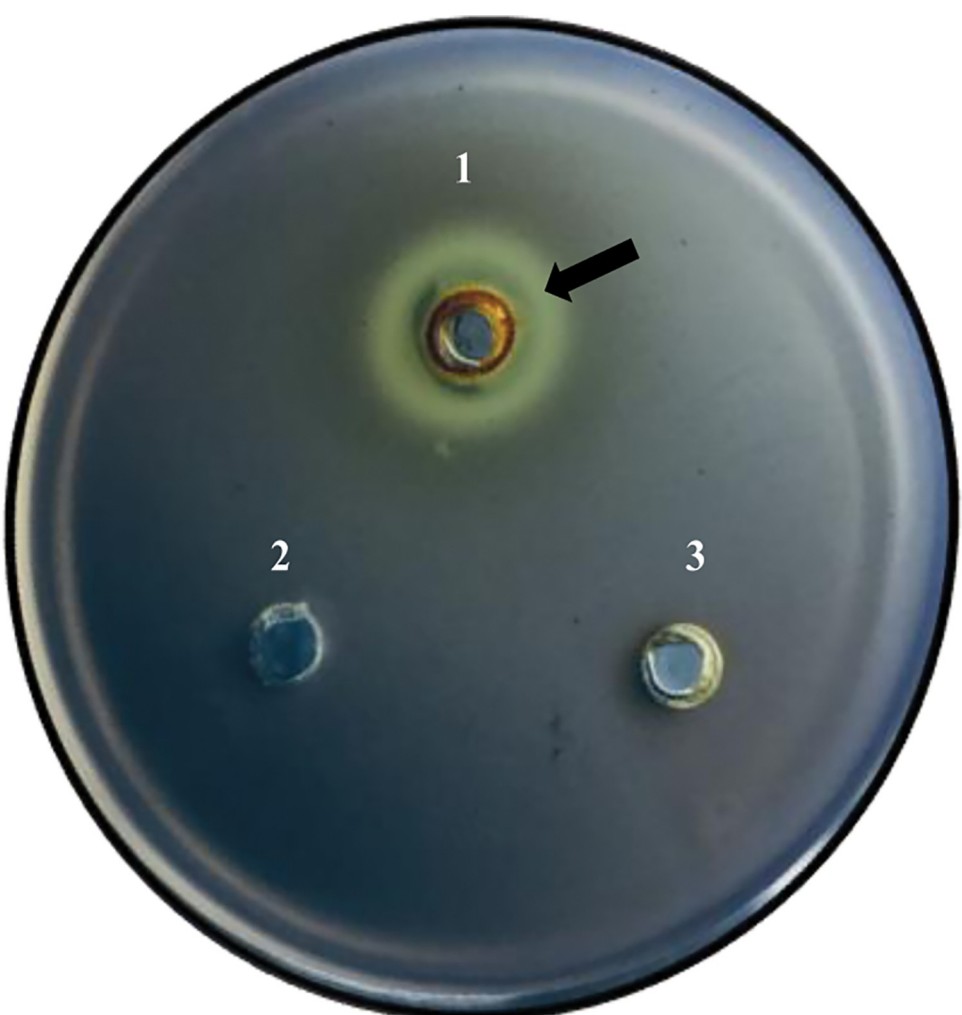

**Fig 2. PGME (well-1) showing anti-quorum sensing activity (arrow indicates colour inhibition); well-2 has DMSO (0.5%) and well-3 has methanol at sub-MIC concentration.**

[41], further emphasizing the efficacy of guava methanolic leaf extract in inhibiting violacein synthesis by approximately 69% in *C. violaceum*.

### Inhibition of virulence factors

The impact of PGME on biofilm development of *C. violaceum* and *P. aeruginosa* PAO1 was assessed, revealing a dose-dependent suppression of biofilm formation (**Fig 4**). The efficacy of PGME was pronounced in reducing PAO1 biofilm formation by 70.07%, 49.60%, 34.64%, and 23.62% at concentrations of 500, 250, 125, and 62.5 μg/mL, respectively, compared to the control. Similarly, *C. violaceum* biofilm formation was inhibited by 59.22% at the highest sub-MIC concentration of PGME (250 μg/mL). Similar outcomes were reported in studies involving *Allium cepa* [42], *Carum copticum* [43], and *Myrtus communis* [44].

The quantitative biofilm data was validated through scanning electron microscopy (SEM). Microscopic evaluation revealed a substantial reduction in the formation of dense mat-like biofilms on coverslip surfaces after treatment with respective sub-MIC concentrations. SEM images (**Fig 5**) illustrated the reduction in bacterial cell aggregation on the glass surface in the

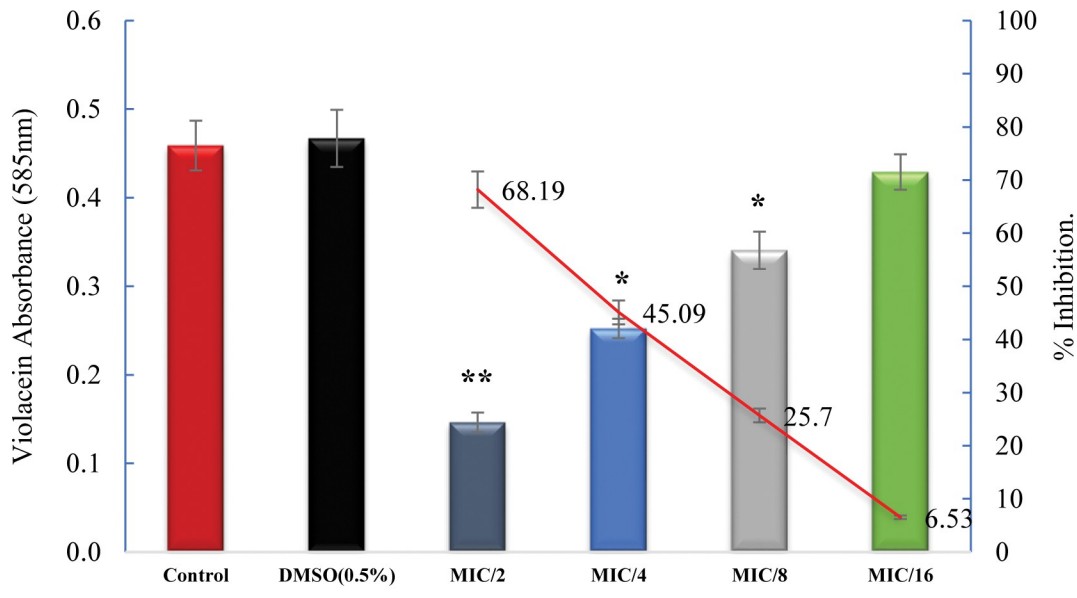

**Fig 3. Quantitative assessment of violacein reduction by PGME in *C. violaceum*.** All data are presented as mean, and the bar represents the standard deviation (SD). * p≤ 0.05 and ** p ≤ 0.01.

presence of PGME. GC/MS analysis further identified several bioactive compounds in the active fraction of *Psidium guajava*, such as caryophyllene, alpha-copaene, and erolidol, known for their antibiofilm properties [45, 46]. Quercetin also detected in PGME, has been previously reported to possess potent antibiofilm activity against pathogenic bacteria [47].

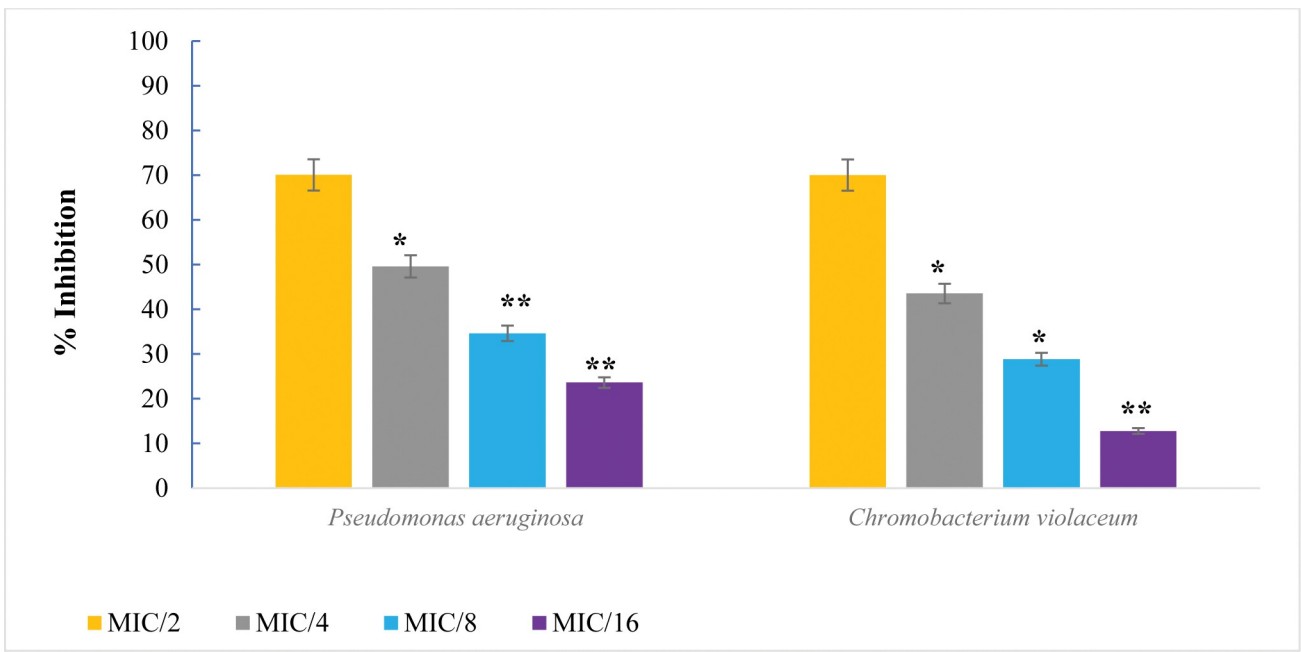

**Fig 4. The effect of PGME sub-MICs on biofilm formation in *C. violaceum* and *P. aeruginosa* PAO1.** The data is shown as the mean of triplicate with the bar representing the standard deviation. *p ≤ 0.05, ** p ≤0.01.

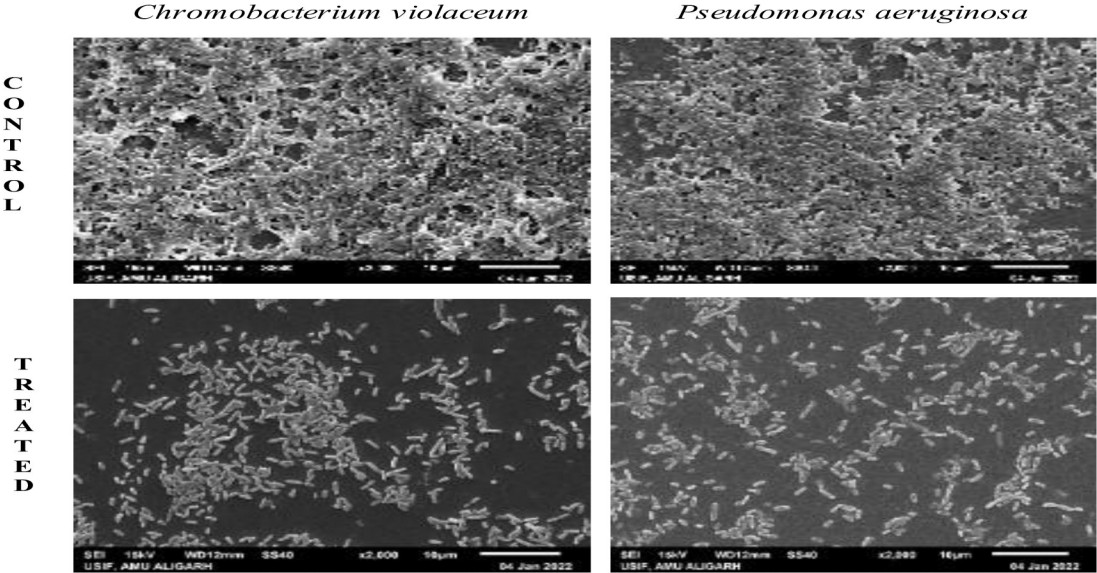

**Fig 5. Scanning electron microscopy images of biofilm of *C. violaceum* and *P. aeruginosa* PAO1.**

Pyocyanin, a virulence factor produced by *P. aeruginosa*, was significantly reduced by PGME treatment (S1 Table in S1 File and **Fig 6**). This pigment induces ROS in host cells by oxidizing reduced glutathione, which correlates with infection severity [48]. At the highest sub-MIC concentration of PGME (500 µg/mL), pyocyanin synthesis was diminished by approximately 70.64%. Additionally, pyocyanin concentration decreased by 42.27%, 26.82%, and 10.11% at concentrations of 250, 125, and 62.5 µg/mL, respectively. Comparable studies involving *Syzygium aromaticum* and *Tinospora cordifolia* demonstrated similar pyocyanin reduction effects [49, 50].

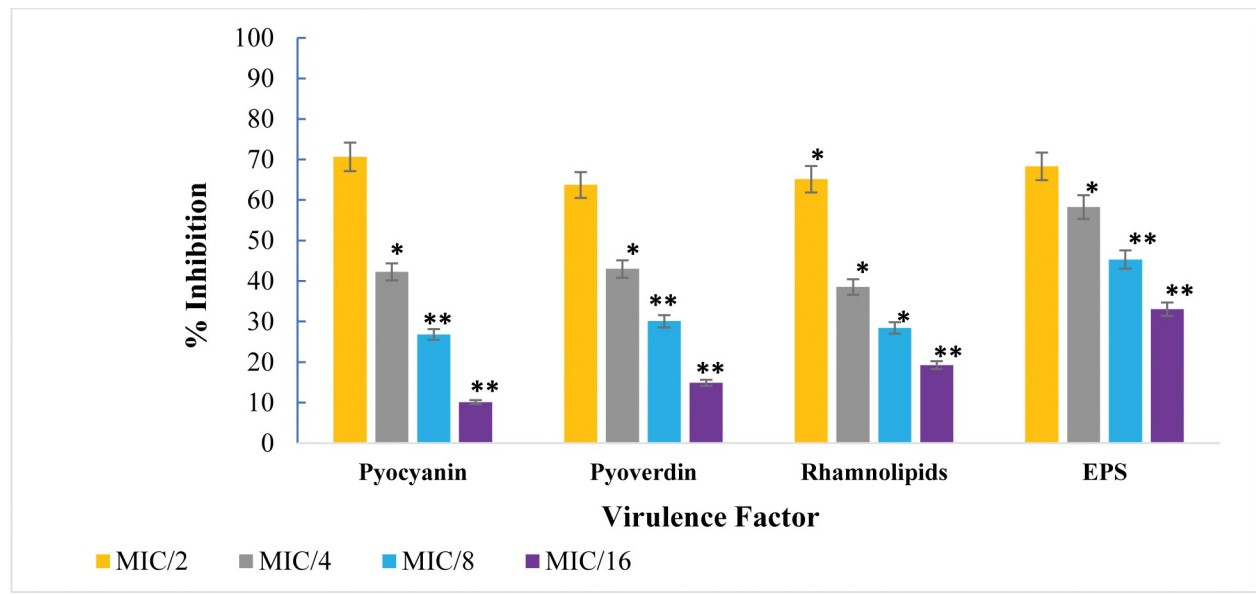

**Fig 6. Virulence factors inhibition in *P. aeruginosa* by sub-MICs of PGME.** The data is shown as the mean of triplicate, bar representing the standard deviation. * $p \leq 0.05$ and ** $p \leq 0.01$.

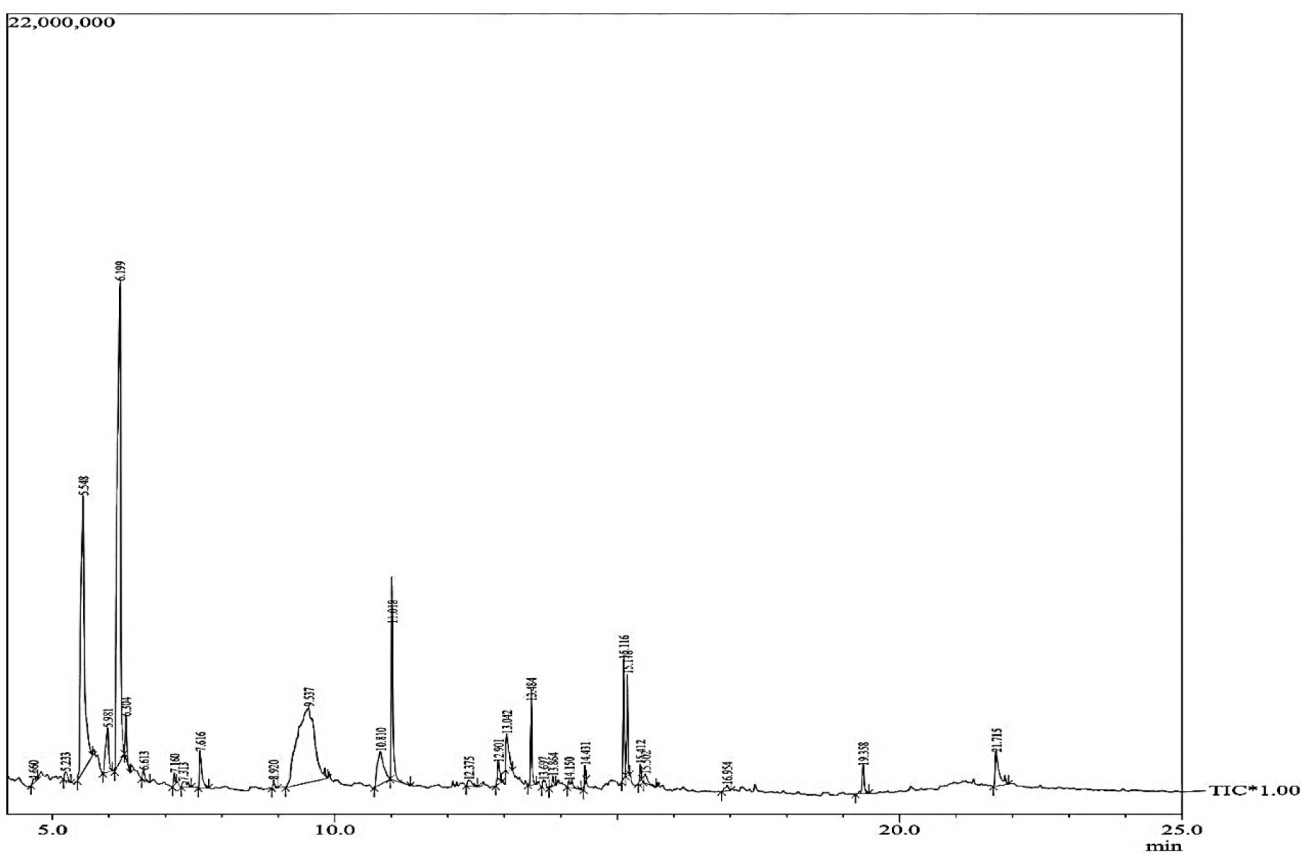

**Fig 7. Gas *chromatography*-mass spectrum chromatogram of PGME extract.**

Pyoverdin, another pigment produced by *P. aeruginosa*, plays a crucial role in enhancing pathogenicity by promoting bacterial growth [51]. The addition of 500 and 250 μg/ml of PGME led to reductions of 63.72% and 42.97%, respectively, in pyoverdin secretion compared to the control. Similarly, *Camellia sinensis* extract reduced pyoverdin production by 58.4% [52].

Rhamnolipids are secreted by *P. aeruginosa* and contribute to biofilm structure maintenance and bacterial adhesion. The presence of PGME at concentrations of 500, 250, 125, and 62.5 μg/mL led to reductions in *P. aeruginosa* PAO1 rhamnolipid synthesis by 65.13%, 38.53%, 28.44%, and 19.26%, respectively. This aligns with the essential role of rhamnolipids in bacterial motility and biofilm formation [53]. Previous research involving *Lagerstroemia speciosa* fruit extract demonstrated a concentration-dependent reduction in pyoverdin production ranging from 3.80% to 51.42% [54].

GC-MS analysis of PGME revealed a diverse profile of phytocompounds (S2 Table in S1 File and **Fig 7**), including significant components such as alpha-copaene, caryophyllene, nerolidol, bisabolene, quercetin, alloaromadendrene, and 9,12-octadecadienoic acid (Z,Z). These results further underline the rich chemical diversity of PGME that contributes to its multifaceted bioactivity.

The results showcase the concentration-dependent efficacy of *Psidium guajava* methanolic extract (PGME) in inhibiting virulence factors and quorum sensing in *C. violaceum* and *P. aeruginosa* PAO1. This comprehensive inhibitory activity underscores the potential of PGME as a valuable natural resource for combating bacterial pathogenicity.

## GC-MS analysis

GC-MS profiling of PGME (S2 Table in S1 File and **Fig 7**) showed different phytocompounds. Specific major components in the profile include alpha-copaene, caryophyllene, nerolidol, bisabolene, quercetin, alloaromadendrene, 9,12-octadecadienoic acid (Z,Z) etc.

## *In silico*: Homology modeling

The 3D structure of the *P. aeruginosa* regulatory protein RhlR has yet to be elucidated. Consequently, the predicted 3D structure generated via the AlphaFold Protein Structure Database was employed for molecular docking studies. The resultant 3D structure of RhlR is depicted in S1A Fig of S1 File. Notably, the AlphaFold algorithm assigns a confidence score (pLDDT) to each residue, where most residues (97.5%) exhibit very high confidence (pLDDT > 90), as indicated by the blue field in S1A Fig of S1 File. A comprehensive analysis of the Ramachandran plot further confirms the accuracy of homology modeling, with 90% of residues residing within the most favored region. The distribution of residues in the most favored, additionally allowed, and generously allowed regions are 92.6%, 7.0%, and 0.5% respectively (S1B Fig in S1 File).

## Molecular docking

Molecular docking studies serve as valuable tools for predicting and exploring potential interactions between compounds and target proteins, including enzymes, RNA, or DNA. The utility of *in silico* molecular docking investigations is particularly notable in assessing the potential pharmacological activities of natural compounds. In this study, molecular docking analysis was conducted using the CB-Dock server to evaluate interactions of the bioactive compounds alpha-copaene, beta-caryophyllene, and nerolidol- identified as key components through GCMS analysis–with *P. aeruginosa* target proteins CviR, LasR, LasI, and RhlR. The CB-dock algorithm operates in three stages, involving the curvature of the protein surface, identification of active site cavities through clustering, and subsequent docking using AutoDock Vina. The superimposition of cocrystal ligands in CviR (PDB ID: 3QP1) and LasR (PDB ID: 2UV0) exhibited re-docking RMSD values of 0.62 Å and 0.78 Å, respectively, confirming the success of the molecular docking study (S2A, S2B Fig in S1 File).

Compounds alpha-copaene, beta-caryophyllene, and nerolidol exhibited favorable binding interactions with target proteins, with interaction energies ranging from -5.5 kcal/mol to -8.6 kcal/mol. The most substantial binding energy interactions were observed against the LasR target protein (S3 Table in S1 File). In-depth analyses of the binding pose of these compounds with the active sites of CviR, LasR, LasI, and RhlR were conducted.

At the CviR active site, the compounds displayed overlapping binding poses. For instance, alpha-copaene engaged in Pi-alkyl and alkyl interactions with residues including Tyr80, Leu57, Ala59, Val75, and Tyr88, alongside van der Waals interactions with Trp111, Ser155, Phe115, Asp97, and Met135. Similarly, beta-caryophyllene formed covalent bonding with Ile99, while nerolidol exhibited Pi-alkyl and alkyl interactions with Tyr88, Pi-sigma interactions with Ile99, Leu57, Leu85, Phe115, Tyr80, Trp84, and Val75, and van der Waals interactions with Ser155, Ala94, and Asp97 (**Fig 8**).

Further binding poses were evident at the LasR active site, where alpha-copaene and beta-caryophyllene established covalent bonds with Val76. Nerolidol, on the other hand, formed a hydrogen bond of 3.88 Å with Tyr47, in addition to Pi-sigma interactions with Trp88. All three compounds exhibited Pi-alkyl and alkyl interactions with Ala70, Ala127, Ile52, and van der Waals interactions with Gly38, Arg61, and Ser129 (**Fig 9**).

Similarly, at the LasI active site, alpha-copaene displayed Pi-alkyl interactions with Val26, Phe27, Trp33, Val148, Phe117, and Ile107, alongside hydrophobic van der Waals interactions

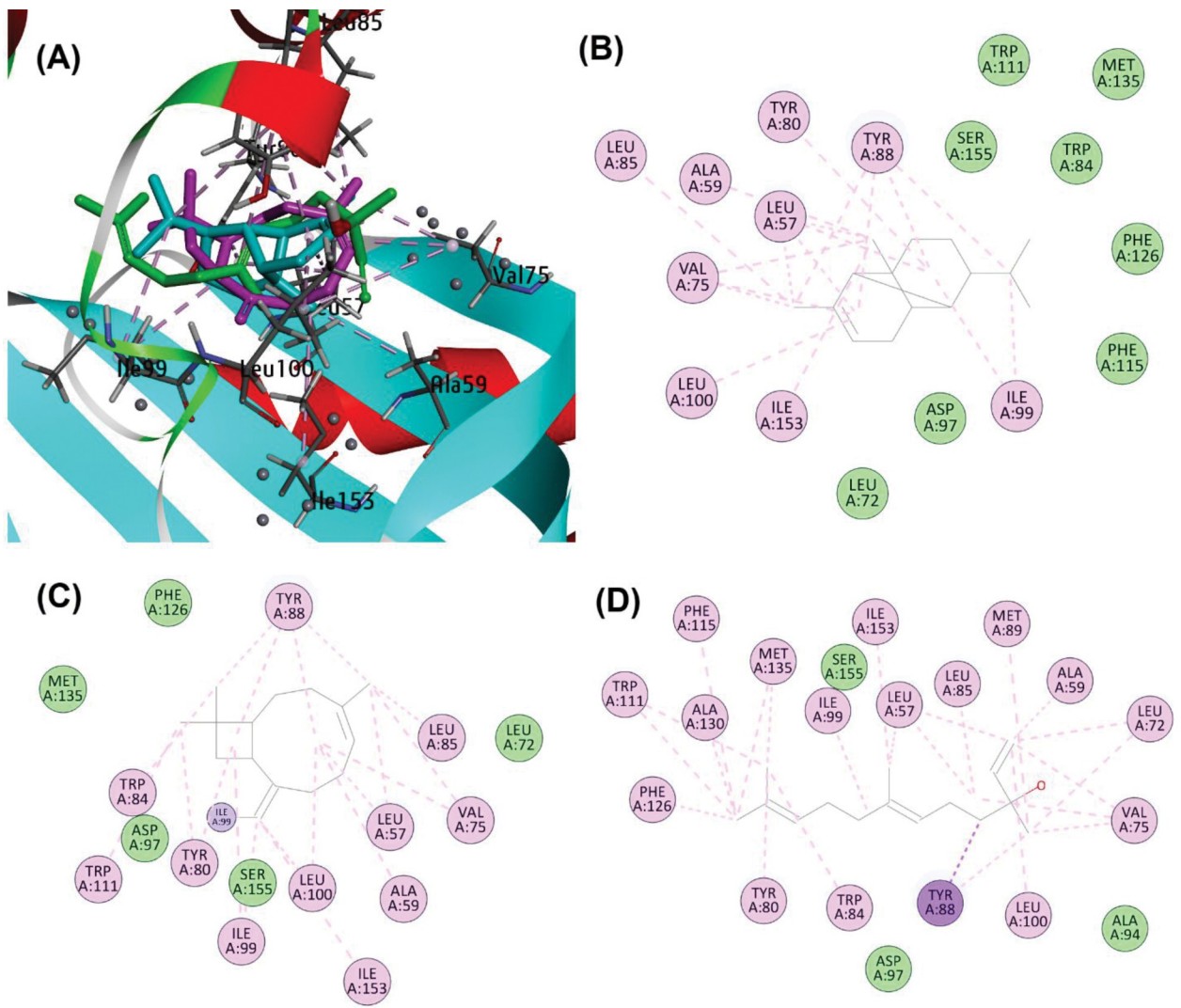

**Fig 8.** (A) Binding poses of alpha-Copaene (cyan), beta-Caryophyllene (magenta) and Nerolidol (green) at active site of CviR (PDB ID: 3QP1). (B) Schematic protein-ligand interaction diagram of alpha-Copaene, (C) beta-Caryophyllene, and (D) Nerolidol.

with Arg30, Thr144, Phe105, Trp69, Thr121, and Ala106. Beta-caryophyllene formed alkyl and Pi-alkyl interactions with Val26, Val148, Phe105, and Trp69, while nerolidol established a covalent bond with Thr145, a hydrogen bond of 4.54 Å with Thr144, and unfavorable metal-donor interactions with Val148 (**Fig 10**).

Lastly, for the RhlR protein, whose 3D structure was constructed through homology modeling, compounds alpha-copaene, beta-caryophyllene, and nerolidol exhibited binding modes involving alkyl and Pi-alkyl interactions with Leu153, Phe7, Arg2, and Phe146. Hydrophobic interactions were noted with Asp4, Asn3, Glu149, and Glu150. Notably, these interactions validate the ability of these compounds to bind at the active sites of target proteins (RhlR, LasI, LasR, and CviR), thus potentially counteracting QS-related trait expression (**Fig 11**).

The collective results demonstrate that alpha-copaene, beta-caryophyllene, and nerolidol exhibit promising binding interactions with key target proteins, suggesting their role in inhibiting quorum sensing-related behaviors.

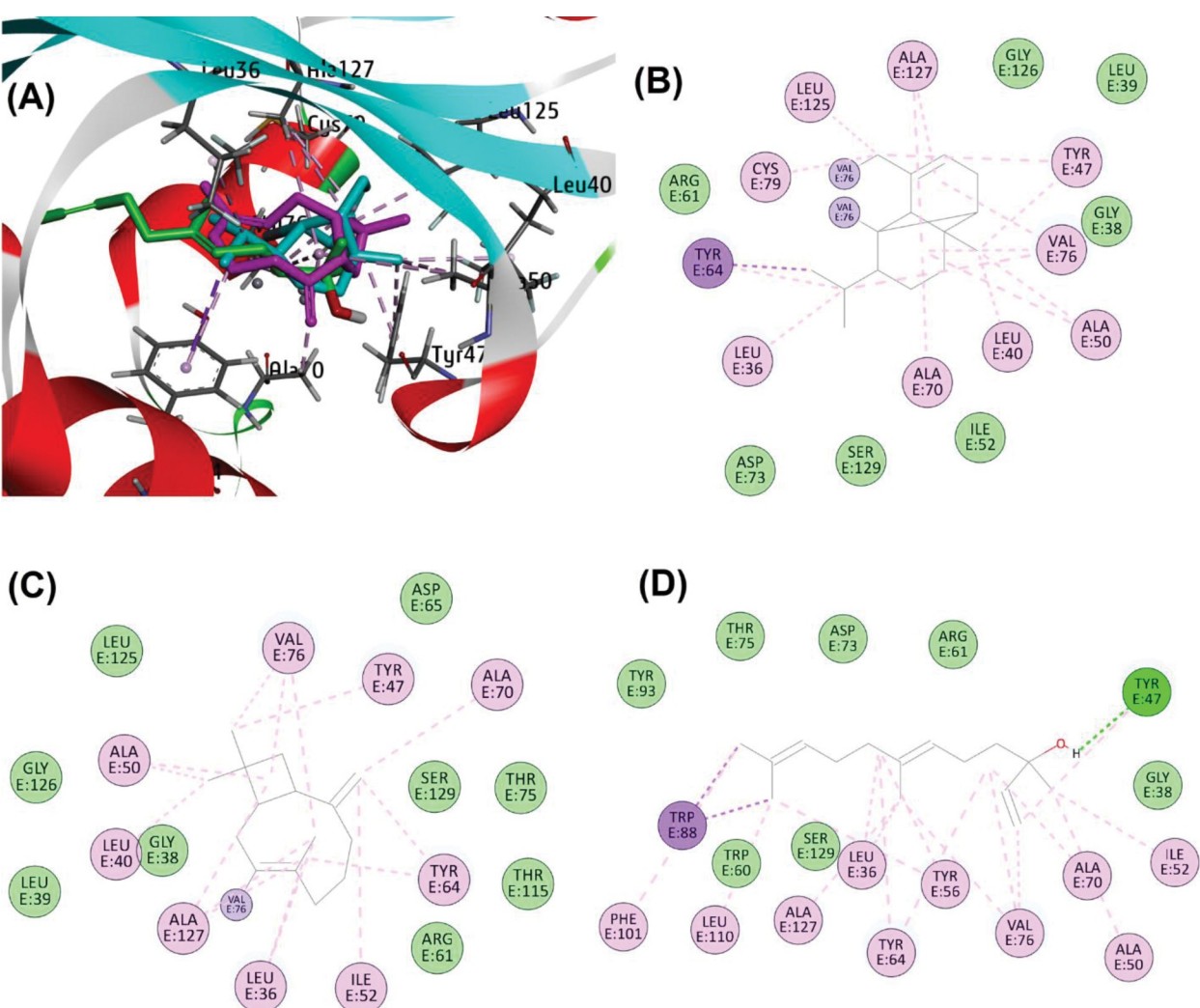

**Fig 9.** (A) Binding poses of alpha-Copaene (cyan), beta-Caryophyllene (magenta) and Nerolidol (green) at active site of LasR (PDB ID: 2UV0). (B) Schematic protein-ligand interaction diagram of alpha-Copaene, (C) beta-Caryophyllene, and (D) Nerolidol.

## Importance of the study

The present study underscores the substantial anti-quorum sensing potential of the methanolic extract of *Psidium guajava* leaves (PGME), highlighting its role in curbing virulence factors like biofilm formation, pyoverdin, pyocyanin, and rhamnolipid synthesis in *P. aeruginosa* PAO1 strain. The uniqueness of this study lies in its comprehensive exploration of QS-controlled anti-virulence activities within a single publication, thereby bridging gaps in the existing fragmented literature. Particularly noteworthy is the dearth of investigations concerning the inhibitory effects of *Psidium guajava* on rhamnolipid production, a crucial factor in biofilm dispersion and motility at infection sites. The GC-MS analysis of PGME unveiled key compounds including alpha-copaene, caryophyllene, and nerolidol. While these compounds have been previously identified in various plant sources, their in-depth *in-silico* docking interactions remain largely unexplored. Our docking analysis illuminates their potential binding and interaction with critical QS-receptors, namely RhlR, CviR, LasI, and LasR proteins,

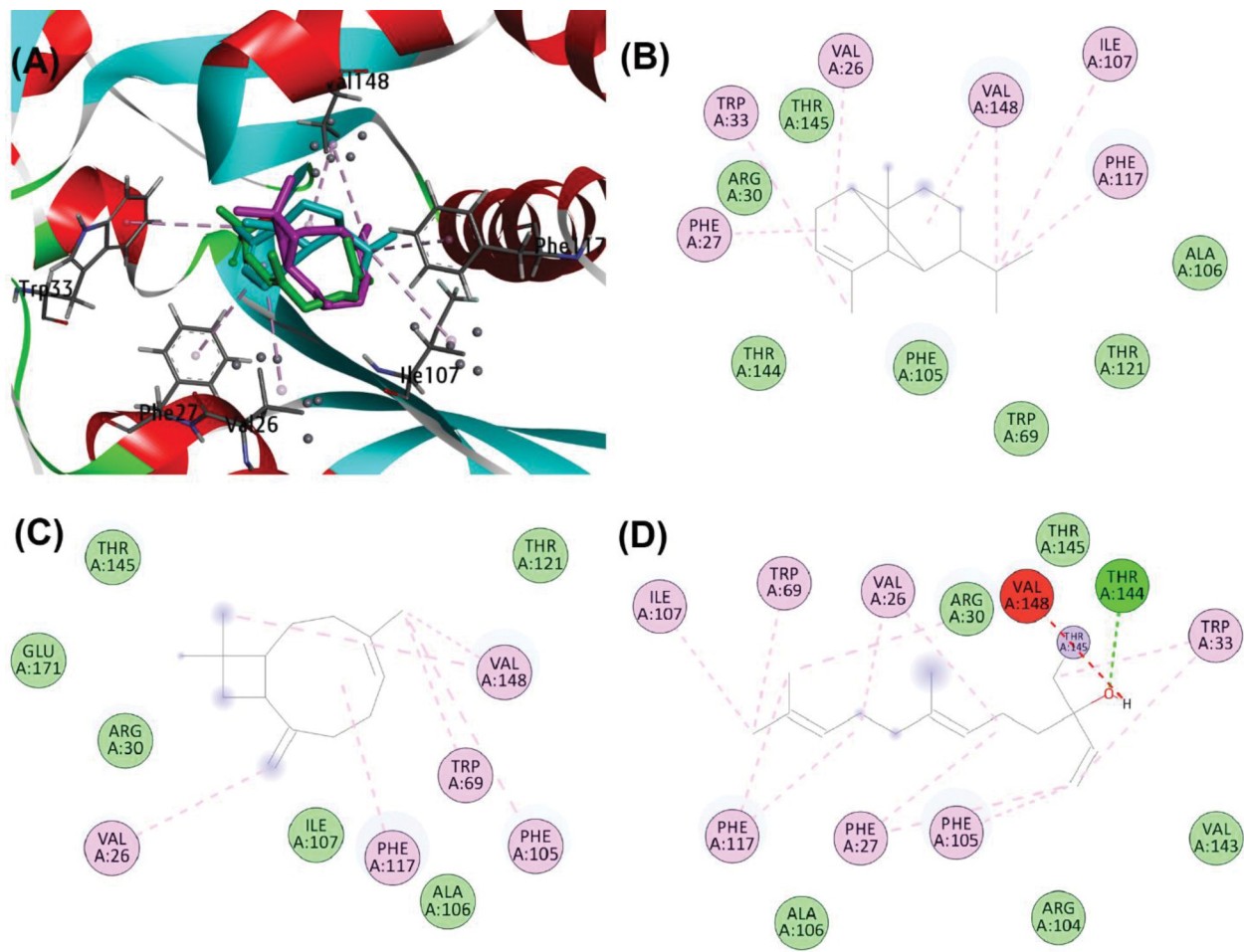

**Fig 10.** (A) Binding poses of alpha-Copaene (cyan), beta-Caryophyllene (magenta) and Nerolidol (green) at active site of LasI (PDB ID: 1RO5). (B) Schematic protein-ligand interaction diagram of alpha-Copaene, (C) beta-Caryophyllene, and (D) Nerolidol.

suggesting potential mechanisms involving AHL suppression and receptor protein obstruction for impeding QS processes. In summation, our study signifies the prospect of PGME as an agent to thwart biofilm formation and quorum sensing, holding promise for combatting drug-resistant bacterial infections. Continued investigation of these attributes might pave the way for innovative treatments targeting such infections.

## Conclusion

This investigation has unequivocally demonstrated that the extract derived from *Psidium guajava* L. exerts a profound reduction in quorum sensing-dependent virulence factor synthesis and biofilm formation, evident across both *P. aeruginosa* and *C. violaceum*. Moreover, this study underscores the pivotal role of employing molecular docking methodologies in uncovering novel therapeutic avenues. The PGME extract has proven to be a potent inhibitor of critical *P. aeruginosa* virulence factors while also disrupting violacein synthesis in *C. violaceum*. The profound potential of the plant extract's active phytocompounds in attenuating QS-related virulence highlights a promising trajectory for future research in the pursuit of antimicrobial agents that mitigate the risks of antibiotic resistance.

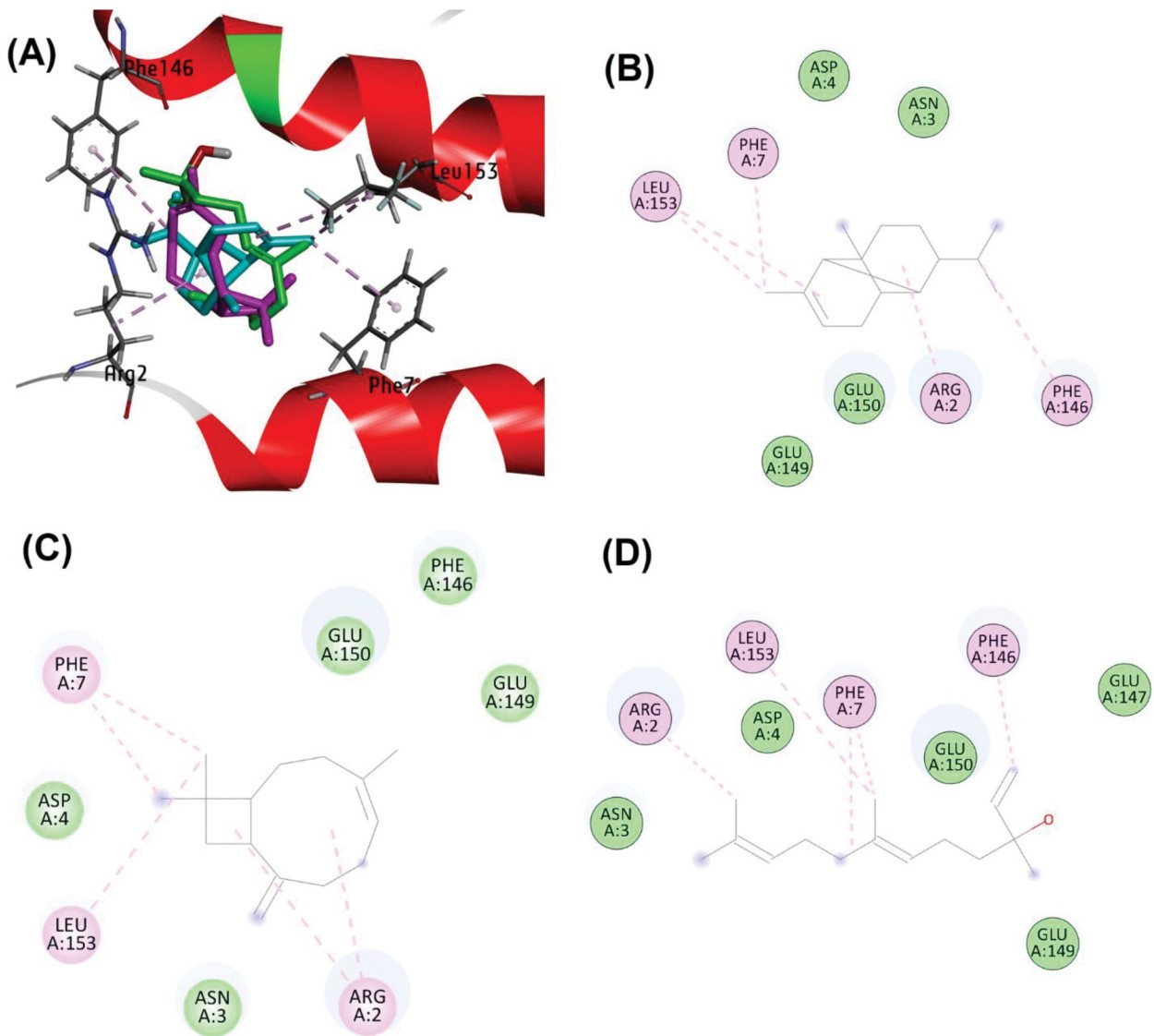

**Fig 11.** (A) Binding poses of alpha-Copaene (cyan), beta-Caryophyllene (magenta) and Nerolidol (green) at active site of RhlR (UniProtKB: P54292). (B) Schematic protein-ligand interaction diagram of alpha-Copaene, (C) beta-Caryophyllene, and (D) Nerolidol.

## Supporting information

**S1 File.**
(DOCX)

## Author Contributions

**Conceptualization:** Mo Ahamad Khan, Haris M. Khan, Mohammad Shahid.

**Data curation:** Mo Ahamad Khan, Sachin Kumar, Bilal Ahmed.

**Formal analysis:** Ismail Celik, Anwar Shahzad.

**Funding acquisition:** Anwar Shahzad.

**Investigation:** Mo Ahamad Khan, Mohammad Shahid, Sachin Kumar, Bilal Ahmed.

**Methodology:** Mo Ahamad Khan, Haris M. Khan, Sachin Kumar.

**Project administration:** Haris M. Khan, Anwar Shahzad.

**Resources:** Mohammad Shahid.

**Software:** Ismail Celik, Mohammad Shahid.

**Supervision:** Ismail Celik, Haris M. Khan, Anwar Shahzad.

**Validation:** Ismail Celik, Haris M. Khan, Sachin Kumar.

**Visualization:** Mo Ahamad Khan, Haris M. Khan, Sachin Kumar, Bilal Ahmed.

**Writing – original draft:** Mo Ahamad Khan, Ismail Celik, Haris M. Khan, Mohammad Shahid, Bilal Ahmed.

**Writing – review & editing:** Mo Ahamad Khan, Ismail Celik, Haris M. Khan, Mohammad Shahid, Anwar Shahzad, Sachin Kumar, Bilal Ahmed.

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
