## [Decision Letter · Decision Letter 0]

21 Aug 2023

PONE-D-23-23418GC-MS profiling, antibiofilm and anti-quorum sensing activity of Psidium guajava L. leaves extract: in vitro and in silico approachPLOS ONE

Dear Dr. Ahmed,

Thank you for submitting your manuscript to PLOS ONE. After careful consideration, we feel that it has merit but does not fully meet PLOS ONE’s publication criteria as it currently stands. Therefore, we invite you to submit a revised version of the manuscript that addresses the points raised during the review process.

We look forward to receiving your revised manuscript.

Kind regards,

M. Alejandro Dinamarca, Dr.

Academic Editor

PLOS ONE

Journal Requirements:

   "No"

Additional Editor Comments:

The work presented to Plos One is part of the search for new control strategies for pathogens, different from the use of antibiotics. In this case, inhibiting the deployment of virulence systems by inhibiting bacterial quorum sensing. Although the work presented is interesting and logical in its structure, it is possible to appreciate details framed in the field of microbiology related to experimental design, methodologies, expression of results and their interpretation, which cast doubt on the scientific conclusions obtained. The English language must be revised

In the work presented, extracts of natural origin of the species Psidium guajava are evaluated to be evaluated against the bacterium Pseudomonas aeruginosa and its virulence mechanisms associated with pathogenicity. The authors design experiences with biosensors for use in this area such as C. violaceum and evaluate the effect of the extracts on the expression of virulence factors such as the expression of EPS, rhamnolipids.

Additionally, depending on the chemical characterization of the active extracts, the authors carry out the exercise of modeling the possible molecular interactions, which is interesting, however, it is not conclusive due to the following critical precedents found in the article:

1. Line 132, method to quantify violacein. The authors state: "Following incubation, the treated culture was centrifuged to isolate the insoluble violacein, which was then suspended in DMSO and centrifuged again to remove the cells. The resulting supernatants were then transferred to microtiter plate wells for the measurement of violacein at 585 nm." Unfortunately, although the methodology is precise in the method reference, there is no detailed experimental protocol that allows determining the relationship between violacein production and cell number or biomass. this is critical and necessary in terms of properly normalizing the data.

2- A similar situation can be observed in the methodology in lines 136-146 in which the Biofilm Inhibition assay method is described. There is no relationship between the measurement of crystal violet color with respect to a relative or normalized value with the bacterial biomass in culture or number of cells.

According to the above, there is a problem related to the data used to conclude the effect of an extract, its characterization and subsequent molecular interaction tests, which may be unconsciously wrong due to a methodological failure.

For these reasons, it is necessary for the authors to review their manuscript and establish whether the results expressed in graphs 2, 3 and 5 are correct or do not represent reality once the data has been normalized with the records of bacterial biomass or number of cells. (turbidity or CFU).

Reviewers' comments:

Reviewer's Responses to Questions

**Comments to the Author**

1. Is the manuscript technically sound, and do the data support the conclusions?

Reviewer #1: Partly

2. Has the statistical analysis been performed appropriately and rigorously? 

Reviewer #1: Yes

3. Have the authors made all data underlying the findings in their manuscript fully available?

Reviewer #1: Yes

4. Is the manuscript presented in an intelligible fashion and written in standard English?

Reviewer #1: No

5. Review Comments to the Author

Reviewer #1: The study by Khan and colleagues reports the anti-quorum sensing and antibiofilm activity of methanolic extract of Psidium guajava leaves. The authors also analyzed the extract's main phytochemicals by GC-MS and described potential interactions of three constituents with some quorum sensing molecules by molecular docking.

-A potential adverse effect of antimicrobials is the selection of resistant microorganisms. Thus, the information that the widespread use of antimicrobials causes resistance should be modified.

-Natural products derived from plants have been used as antimicrobials since ancient times. Researchers who have renewed interest in studying these products.

-Virulence factors are essential for bacterial survival in certain situations. For example, during an infectious process. Biofilms are the most common mode of microbial growth in different environments.

-According to the authors, one of the objectives of the study is to evaluate the effect of P. guajava extract on the formation of virulence factors. The methodology used is not adequate to meet this objective, since the inhibitory effect on the expression of the selected virulence factor was analyzed.

-If one of the objectives is to select substances/extracts with potential anti-quorum sensing or anti-biofilm activity, why was only the methanolic extract, which showed growth inhibition activity, selected? Have the other extracts been tested on biofilms?

-What quality controls of the tests: substance with known inhibitory effect in relation to the test performed with the extract? Microdilution test control bacteria for MIC determination?

-In the violacein quantification test, a numerical value is expected for the results, and not as optical density values.

-The cell density of the inoculum for the biofilm assay must be reported in order for the result to be reproduced.

-Biofilms were treated with which extract concentration for electron microscopy analyses? The methodology needs to be better described.

-The authors describe the results and compare them with other results obtained with different plants. However, this comparison must be done carefully, taking into account the plant species, the type of extract and the main chemical constituents.

-The English language must be revised.

-Line 263: What does "aggerate-like structure" mean

-Line 267: “EPS is a major virulence factors” please explain.

6. PLOS authors have the option to publish the peer review history of their article (what does this mean?). If published, this will include your full peer review and any attached files.

Reviewer #1: No

---

## [Author Response · Author response to Decision Letter 0]

13 Sep 2023

A 'response to reviewers' file has been uploaded.

---

## [Decision Letter · Decision Letter 1]

12 Oct 2023

PONE-D-23-23418R1GC-MS profiling, antibiofilm and anti-quorum sensing activity of Psidium guajava L. leaves extract: in vitro and in silico approachPLOS ONE

Dear Dr. Ahmed,

Thank you for submitting your manuscript to PLOS ONE. After careful consideration, we feel that it has merit but does not fully meet PLOS ONE’s publication criteria as it currently stands. Therefore, we invite you to submit a revised version of the manuscript that addresses the points raised during the review process.

The main concern: please show the effect on the bacterial growth at sub-MIC concentration

*Line 125 “Antimicrobial screening and *
*minimal inhibitory concentration (MIC) determination”*

*The authors do not present methodology information that allows supporting the work with known concentrations or proportions of the evaluated PGME extract. The authors point out that serial dilutions of the extract were made without indicating whether it was from the original stock solution or a dilution of the stock solution. Considering that the work refers to the evaluation of an extract in its biological activities such as ant virulence or antibiofilm, not having this information is critical for the quality of the paper.*

Line 131 *The authors point out "to generate a concentration range". Again, not knowing the concentration of the extract of the initial stock solution or matrix is critical when indicating the "concentration range". The authors must indicate the original concentration of the extract or its percentage working ratio.*

*Line 139 “**varying concentrations of the fractions were placed on the agar plate” *The authors refer to a neither detailed nor defined concentration of a fraction of the extract containing molecules evaluated.

Line 156 “Biofilm Inhibition Assay” The authors point out *“containing sub-MIC levels of PGME”**. *The authors point out values that do not correspond to concentrations or proportions (percentages) that allow the MIC to be scientifically defined by concentration or proportion.

*The authors point out "to generate a concentration range". In the same line of argument, not knowing the concentration of the extract of the initial stock solution or matrix is critical when indicating "range of concentrations." The authors must indicate the original concentration of the extract, or its percentage working ratio.*

*The authors do not present methodology information that allows supporting the work with known concentrations or proportions of the evaluated extract. The authors point out that serial dilutions of the extract were made without indicating whether it was from the original stock solution or a dilution of the stock solution. Considering that the work refers to the evaluation of an extract in its biological activities such as antivirulence or antibiofilm, not having this information is critical for the quality of the paper.*

*Finally, a* normalization of CFU data is presented to support previous observation response in relation to the effect of bacterial growth or biomass with respect to the extract concentrations used. The normalization of CFU data by applying logarithm does not allow us to show the differences in biomass that in a range of 1 logarithm can be significant. This, if not considered in the effect of measuring the evaluated activity, can generate artifacts associated with data analysis. The authors must present data on the effects evaluated in relation to the growth of the bacteria evaluated and concentrations or proportions of the evaluated PGME extract.

We look forward to receiving your revised manuscript.

Kind regards,

M. Alejandro Dinamarca, Dr.

Academic Editor

PLOS ONE

Reviewers' comments:

Reviewer's Responses to Questions

**Comments to the Author**

1. If the authors have adequately addressed your comments raised in a previous round of review and you feel that this manuscript is now acceptable for publication, you may indicate that here to bypass the “Comments to the Author” section, enter your conflict of interest statement in the “Confidential to Editor” section, and submit your "Accept" recommendation.

Reviewer #2: All comments have been addressed

Reviewer #3: All comments have been addressed

2. Is the manuscript technically sound, and do the data support the conclusions?

Reviewer #2: Partly

Reviewer #3: Yes

3. Has the statistical analysis been performed appropriately and rigorously? 

Reviewer #2: Yes

Reviewer #3: Yes

4. Have the authors made all data underlying the findings in their manuscript fully available?

Reviewer #2: Yes

Reviewer #3: Yes

5. Is the manuscript presented in an intelligible fashion and written in standard English?

Reviewer #2: Yes

Reviewer #3: Yes

6. Review Comments to the Author

Reviewer #2: The authors have adequetly addressed tha reviewer comments but the title should be modified and the introduction should be summarised. Also, Figure showing the inhibitory effect of the extract on the virulence factors should be added.

More recent references should be cited like:

Alotaibi B, Negm WA, Elekhnawy E, El-Masry TA, Elseady WS, Saleh A, Alotaibi KN, El-Sherbeni SA. Antibacterial, immunomodulatory, and lung protective effects of Boswellia dalzielii oleoresin ethanol extract in pulmonary diseases: in vitro and in vivo studies. Antibiotics. 2021 Nov 25;10(12):1444.

Attallah NG, El-Kadem AH, Negm WA, Elekhnawy E, El-Masry TA, Elmongy EI, Altwaijry N, Alanazi AS, Al-Hamoud GA, Ragab AE. Promising antiviral activity of Agrimonia pilosa phytochemicals against severe acute respiratory syndrome coronavirus 2 supported with in vivo mice study. Pharmaceuticals. 2021 Dec 16;14(12):1313.

Attallah NG, El-Sherbeni SA, El-Kadem AH, Elekhnawy E, El-Masry TA, Elmongy EI, Altwaijry N, Negm WA. elucidation of the metabolite profile of Yucca gigantea and assessment of its cytotoxic, antimicrobial, and anti-inflammatory activities. Molecules. 2022 Feb 16;27(4):1329.

Reviewer #3: The current study evaluates the anti-QS and anti-virulence activities of guava leaves against Pseudomonas aeruginosa. The study is well designed and conclusive.

The mani concern: please show the effect on the bacterial growth at sub-MIC concentration

Minors:

- Please write the names of genes in italic

- Add more information about the advantages of employing this strategy in conquering bacterial resistance, please find helpful publications

https://doi.org/10.3390/ijms232113088

https://doi.org/10.3390/microorganisms10101976

https://doi.org/10.3390/app11156847

https://doi.org/10.3390/biomedicines10051169

- Furthermore, if there are previous studies exploring the guava leaves antibacterial and anti-virulence activities, please include in your results and discussion

7. PLOS authors have the option to publish the peer review history of their article (what does this mean?). If published, this will include your full peer review and any attached files.

Reviewer #2: No

Reviewer #3: No

---

## [Author Response · Author response to Decision Letter 1]

6 Nov 2023

Response to all the comments raised by the editor and reviewers can be found in the word file.

---

## [Editor Report · Decision Letter 2]

22 Nov 2023

Antibiofilm and anti-quorum sensing activity of Psidium guajava L. leaf extract: in vitro and in silico approach

PONE-D-23-23418R2

Dear Dr. Bilal Ahmed,

We’re pleased to inform you that your manuscript has been judged scientifically suitable for publication and will be formally accepted for publication once it meets all outstanding technical requirements.

Kind regards,

M. Alejandro Dinamarca, Dr.

Academic Editor

PLOS ONE
---

## [Editor Report · Acceptance letter]

8 Dec 2023

PONE-D-23-23418R2 

Antibiofilm and anti-quorum sensing activity of Psidium guajava L. leaf extract: in vitro and in silico approach 

Dear Dr. Ahmed:

I'm pleased to inform you that your manuscript has been deemed suitable for publication in PLOS ONE. Congratulations! Your manuscript is now with our production department. 

Kind regards, 

on behalf of

Mr M. Alejandro Dinamarca 

Academic Editor

PLOS ONE